

# Relationship among airborne pollen, sensitization, and pollen food allergy syndrome in Asian allergic children

Yoonha Hwang[1], Chikako Motomura[2], Hironobu Fukuda[3], Reiko Kishikawa[4], Naoto Watanabe[5] and Shigemi Yoshihara[3]

[1] Department of Pediatrics, Busan St. Mary's Hospital, Busan, South Korea
[2] Department of Pediatrics, National Hospital Organization Fukuoka National Hospital, Fukuoka, Japan
[3] Department of Pediatrics, Dokkyo Medical University Hospital, Tochigi, Japan
[4] Department of Allergy, National Hospital Organization Fukuoka National Hospital, Fukuoka, Japan
[5] Department of Allergy Internal Medicine, Seirei Yokohama Hospital, Kanagawa, Japan

## ABSTRACT

**Background.** Causes of pediatric pollen food allergy syndrome (PFAS) differ depending on airborne pollen levels in a particular region. We aimed to analyze airborne pollen counts, IgE sensitization rates, and PFAS incidence among children with allergies in South Korea and Japan.

**Methods.** This cross-sectional study included children aged 5–17 years with allergies in 2017. Airborne pollen samples were collected from Busan in South Korea, and Fukuoka and Tochigi in Japan. Questionnaires were used to assess bronchial asthma, seasonal allergic rhinitis, atopic dermatitis, food allergy, and PFAS. The serum IgE specific to Dermatophagoides pteronyssinus, pollen, tomato, and peach were investigated.

**Results.** In total, 57, 56, and 20 patients from Busan, Fukuoka, and Tochigi, respectively, were enrolled. Airborne Japanese cedar and cypress pollen were predominant in Fukuoka and Tochigi, whereas pine and alder pollen were predominant in Busan. Children with allergies in Fukuoka and Tochigi had a significantly higher sensitization rate to Japanese cedar, cypress, juniper, orchard grass, ragweed, Japanese hop, and tomato compared with children in Busan. In Fukuoka and Tochigi, where Japanese cedar and cypress pollen were frequently scattered, high sensitizations among allergic children were observed. The sensitization rate was not affected by the pollen count in alder, grass, ragweed, and Japanese hop. In multivariable analysis, only alder sensitization was found to be associated with PFAS (odds ratio: 6.62, 95% confidence interval: 1.63–26.87, $p = 0.008$). In patients with PFAS in Busan and Tochigi, peach associated with birch allergen Bet v 1 was a causative food item for PFAS. Moreover, PFAS was associated with ragweed and Japanese hop pollen sensitization in Fukuoka.

**Conclusion.** Regardless of pollen counts, alder pollen sensitization was associated with PFAS in children. Ragweed and Japanese hop pollen sensitization were associated with PFAS, particularly among children in southern Japan.

Corresponding author
Chikako Motomura, motomura.chikako.wv@mail.hosp.go.jp

## INTRODUCTION

Airborne pollen from birch, alder, hazel, oak, hornbeam, chestnut, and beech are predominant in Northern and Central Europe and are a major cause of allergic rhinitis and possibly asthma symptoms (*Biedermann et al., 2019*). The broad cross-reactivity of birch pollen allergens extends to plant food allergens, resulting in pollen food allergy syndrome (PFAS). PFAS has different causes depending on the major pollen present in the region. The prevalence of PFAS is reportedly varied, ranging from 4.7% to >20% in children and 13%–58% in adults (*Carlson & Coop, 2019*). In Japan, the primary causative foods for initial food allergy are fruits among 4–6-year-olds and crustaceans (first) and fruits (second) among 7–19-year-olds (*Ebisawa, Ito & Fujisawa, 2020*). Few reports have assessed pediatric PFAS in East Asian countries such as Japan and South Korea (*Takemura et al., 2020*; *Kim et al., 2018*).

Busan (South Korea), Fukuoka and Tochigi (Japan) have similar average temperature, humidity, and types of fruits ingested; however, they differ in terms of tree vegetation (Fig. 1). In Japan, airborne pollen from Japanese cedar and other related trees are predominant and are major causes of seasonal allergic rhinitis (AR) (*Kishikawa et al., 2017*). In contrast, several pine pollen are scattered throughout South Korea. The airborne pollen counts may affect pollen sensitization, which may affect the prevalence of PFAS in the area. Therefore, the present study aimed to assess airborne pollen counts, Immunoglobulin E (IgE) sensitization rates, and PFAS prevalence among children with allergies in three regions of South Korea and Japan.

## MATERIALS & METHODS

This multicenter prospective cross-sectional study was conducted at the Department of Pediatrics, Busan St. Mary's Hospital, National Hospital Organization Fukuoka National Hospital and Dokkyo Medical University Hospital in Tochigi. Of the outpatients aged 5–17 years who visited three different institutions in 2017 for allergic diseases (bronchial asthma (BA), allergic rhinitis (AR), food allergy (FA), or atopic dermatitis (AD)), 57 out of 169 (34%) in Busan, 56 out of 384 (15%) in Fukuoka, and 20 out of 222 (9%) in Tochigi agreed to participate in the survey. A limiting factor for enrollment in this study would be if the patient had a recent blood test.

### Diagnostic criteria and procedures

The translated PFAS questionnaire was used to assess PFAS, its causative food items, and food allergy (FA) (Table S1, *Skypala et al., 2011*). The Japanese and Korean versions of the International Study of Asthma and Allergies in Childhood (ISAAC) Questionnaire were used to determine whether the patients had BA, seasonal AR, or AD (*Rutter et al., 2020*). Allergic diseases, including PFAS, have not been confirmed by physicians. The authors have permission to use these instruments from the copyright holders. Table S2 presents the definitions of allergic diseases.

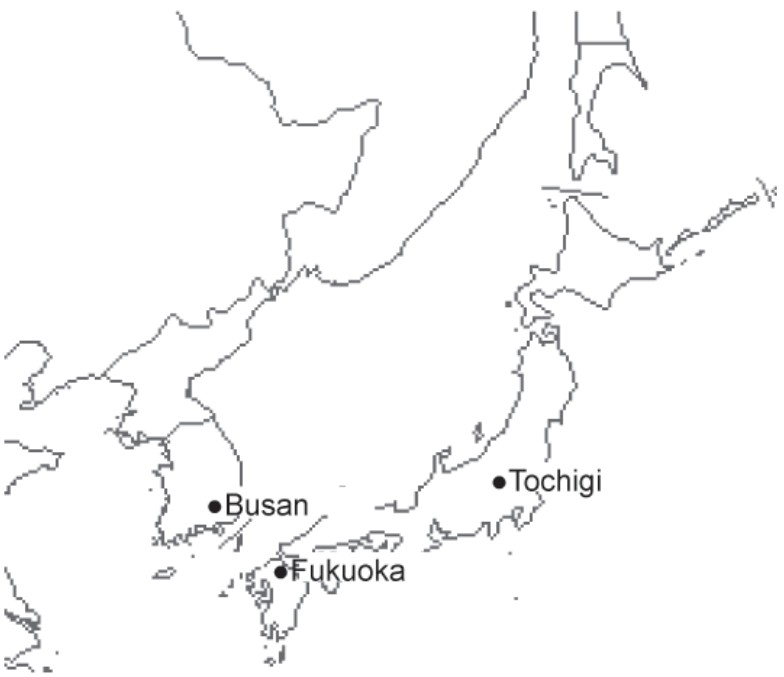

**Figure 1** **Geographical location of Busan, Fukuoka and Tochigi.**

## Annual airborne pollen counts

In Korea, pollen count data are typically collected using a Rotorod collector. In Japan, they are collected using the gravity method. Therefore, two different measurement methods were used in this study.

In Busan, airborne pollen samples were collected by installing the Rotorod sampler (Sampling Technologies, Inc., Minnetonka, MN, USA) on the roof of Busan St. Mary's Hospital. Rotorod collector rods were coated with the standard silicone grease adhesive. After exposure, the rods were stained with Calberla's solution, and mounted on a specially designed microscope stage adapter according to the manufacturer's instructions (*Frenz, 2000*). Samples from all three samplers were assayed under a light microscope at 400× magnification and counts were converted to concentrations of pollen grains per m$^3$ by dividing the number of pollen grains by the volume of air sampled.

Tochigi did not have an observation post for direct evaluation of airborne pollen; therefore, we used the airborne pollen counts of Takasaki, Gunma Prefecture, because it is geographically 55 km close to Tochigi. The results for Fukuoka were obtained directly. Airborne pollen samples at Fukuoka and Tochigi were collected using the Durham's sampler (Nishiseiki, Inc., Funabashi, Chiba, Japan) according to the gravity method. After installing the collector on the building's roof, a thin layer of petroleum jelly was applied on the slide glass, which was exchanged and stained with Calberla's fuchsin dye each morning. Subsequently, the type of airborne pollen was analyzed at a magnification of 400×. The number of pollen grains at a magnification of 100× was calculated and the number of pollen grains per cm$^2$ was recorded.

**Table 1  Dermographics of the study subjects.**

|  | Whole population | Busan | Fukuoka | Tochigi | *p*-value |
|---|---|---|---|---|---|
| Number | 133 | 57 | 56 | 20 |  |
| Mean age (years, range) | 9.5 (5–17) | 9.3 (5–17) | 9.6 (6–14) | 9.8 (6–15) | 0.72 |
| Male (%) | 71 (53) | 26 (46) | 35 (63) | 10 (50%) | 0.18 |
| PFAS (%) | 27 (20) | 7 (12) | 14 (25) | 6 (30) | 0.12 |
| FA (%) | 72 (54) | 14 (25) | 45 (80) | 13 (65) | <0.001 |
| BA (%) | 59 (44) | 16 (28) | 31 (55) | 12 (60) | 0.009 |
| Seasonal AR (%) | 75 (56) | 32 (56) | 32 (57) | 11 (55) | 0.99 |
| AD (%) | 58 (43) | 24 (42) | 27 (48) | 7 (35) | 0.57 |

Notes.
  Abbreviations: AR, allergic rhinitis; AD, atopic dermatitis; BA, bronchial asthma; FA, food allergy; PFAS, pollen food allergy syndrome.

## Pollen and fruit sensitization

ImmunoCAP (Phadia AB, Uppsala, Sweden) tests were performed to assess specific IgE to Dermatophagoides pteronyssinus (Dp), Japanese cedar, cypress, juniper, pine, birch, alder, orchard grass, ragweed, Japanese hop, tomato, and peach. Pollen and fruit sensitizations were defined based on positive specific IgE test result ($\geq$ 0.70 UA/mL).

## Statistical analysis

The Kruskal–Wallis and Chi-squared tests were used to compare numbers and ratios, respectively. A multivariable logistic regression analysis was performed to analyze the effect of several independent variables (location, Japanese cedar, pine, alder, ragweed, and Japanese hop sensitizations). A *p*-value of <0.05 was considered statistically significant. All statistical analyses were completed using Statistical Package for Social Sciences software, version 25.0 (SPSS Inc., Chicago, IL, USA).

## Ethics statement

This research study was conducted after obtaining approval from the Research Ethics Committee of Busan St. Mary's Hospital (BSM 2016-14), Fukuoka National Hospital (F28-3), and Dokkyo Medical University Hospital (29031). Written informed consent was obtained from the study participants and their caregivers.

## RESULTS

In total, 133 patients (Busan: $n = 57$, Fukuoka: $n = 56$, and Tochigi: $n = 20$), were included in the study. There was no significant difference in terms of age, sex, PFAS incidence, seasonal AR, and AD in pediatric patients with allergies among the three sites (Table 1). However, the proportion of participants with BA and FA were higher in Fukuoka and Tochigi than in Busan (BA: 55%, 60%, 28%: $p = 0.009$, FA: 80%, 65%, 25%: $p < 0.001$).

According to the 2017 annual airborne pollen counts of each region (Table 2), in Busan, the pine pollen count was the highest (74%), followed by alder (14%), Japanese hop (7%), juniper (3%), grass (1%), and birch (1%), while airborne cypress pollen (68%) in Fukuoka and Japanese cedar pollen (79%) in Tochigi were predominant. The alder pollen counts in

**Table 2   Annual pollen counts in Busan, Fukuoka and Tochigi in 2017 (per year).**

| Families, common name and genus | Busan/m³ | Fukuoka/cm² | Tochigi/cm² |
|---|---|---|---|
| Family Cupressaceae | | | |
| Japanese cedar, Cryptomeria japonica (%) | 0 | 1852 (26) | 6989 (79) |
| Cypress, Cupressus (%) | 0 | 4844 (68) | 563 (6) |
| Juniper, Juniperus (%) | 260 (3) | 0 | 0 |
| Family Pinaceae | | | |
| Pine, Pinus (%) | 6438 (74) | 289 (4) | 948 (11) |
| Family Betulaceae | | | |
| Birch, Betula (%) | 46 (1) | 0 | 16 (0) |
| Alder, Alnus (%) | 1259 (14) | 1 (0) | 3 (0) |
| Grass (%) | 69 (1) | 45 (1) | 150 (2) |
| Family Asteraceae | | | |
| Ragweed, Ambrosia (%) | 31 (0) | 19 (0) | 55 (1) |
| Family Cannabaceae | | | |
| Japanese hop, Humulus japonicus (%) | 592 (7) | 30 (1) | 112 (1) |
| Total (%) | 8002 (100) | 7080 (100) | 8519 (100) |

Fukuoka and Tochigi were not clearly evaluated, and grass and ragweed pollen counts did not differ among the three sites.

Children in Fukuoka and Tochigi had significantly higher sensitization rates to Japanese cedar ($p < 0.001$), cypress ($p < 0.001$), juniper ($p < 0.001$), orchard grass ($p = 0.007$), ragweed ($p = 0.005$), Japanese hop ($p = 0.005$), and tomato ($p = 0.006$), compared with children in Busan (Table 3). The sensitization rates to Dp, pine, birch, alder, and peach did not differ among children in the three sites. In Fukuoka and Tochigi, where Japanese cedar and cypress pollen were frequently scattered, a high sensitization rate among allergic children was observed. The sensitization rate was not affected by the pollen count in alder, grass, ragweed, and Japanese hop.

Table 3 shows the differences in sensitization rates between children with and without PFAS. In all three sites, the sensitization rates to birch and alder among patients with PFAS were higher than those among patients without PFAS. Additionally, there were regional differences in the sensitization rates between children with and without PFAS; in Busan, patients with PFAS had a greater sensitization rate to Japanese cedar and peach than those without PFAS, whereas in Fukuoka, sensitization rates to pine, ragweed, Japanese hop, and peach were higher in patients with PFAS than in those without PFAS.

Univariable logistic regression analysis showed that Japanese cedar, ragweed, alder, pine, and Japanese hop were statistically significant variables (Table 4). Adolescence was believed to increase PFAS incidence, but older age ($\geq 10$ years) had no significant effect on PFAS. The multivariable logistic regression analysis conducted using PFAS as the dependent variable and location and airborne pollen sensitization as independent variables, showed that only alder sensitization was significantly associated with PFAS in all patients (odds ratio: 6.62, 95% confidence interval: 1.63–26.87, $p = 0.008$)

**Table 3  Differences of sensitization expressed as percentages, with or without pollen food allergy syndrome (PFAS) between 3 sites.**

| | Busan | | | | Fukuoka | | | | Tochigi | | | |
|---|---|---|---|---|---|---|---|---|---|---|---|---|
| | Total (%) n = 57 | PFAS+ (%) n = 7 | PFAS- (%) n = 50 | p | Total (%) n = 56 | PFAS+ (%) n = 14 | PFAS- (%) n = 42 | p | Total (%) n = 20 | PFAS+ (%) n = 6 | PFAS- (%) n = 14 | p |
| Dp | 84 | 86 | 82 | 1.0 | 91 | 93 | 90 | 1.0 | 85 | 83 | 79 | 1.0 |
| Japanese cedar** | 19 | 43 | 6 | 0.020 | 95 | 93 | 88 | 1.0 | 75 | 83 | 71 | 1.0 |
| Cypress** | 8 | 17 | 5 | 0.33 | 84 | 86 | 76 | 0.71 | 70 | 83 | 57 | 0.35 |
| Juniper** | 18 | 29 | 6 | 0.11 | 81 | 80 | 79 | 1.0 | 70 | 83 | 50 | 0.32 |
| Pine | 11 | 14 | 4 | 0.33 | 24 | 44 | 9 | 0.028 | 30 | 33 | 7 | 0.20 |
| Birch | 35 | 71 | 22 | 0.020 | 43 | 55 | 16 | 0.016 | 40 | 83 | 21 | 0.018 |
| Alder | 35 | 71 | 24 | 0.015 | 37 | 57 | 12 | 0.002 | 45 | 83 | 21 | 0.018 |
| Orchard grass* | 21 | 29 | 8 | 0.33 | 46 | 57 | 29 | 0.105 | 50 | 67 | 29 | 0.15 |
| Ragweed* | 11 | 14 | 4 | 0.15 | 35 | 50 | 14 | 0.011 | 32 | 40 | 7 | 0.16 |
| Japanese hop* | 11 | 14 | 8 | 0.51 | 30 | 43 | 5 | 0.002 | 25 | 33 | 14 | 0.55 |
| Tomato* | 15 | 25 | 7 | 0.31 | 42 | 50 | 21 | 0.071 | 35 | 50 | 14 | 0.13 |
| Peach | 21 | 75 | 13 | 0.022 | 31 | 44 | 11 | 0.049 | 35 | 50 | 21 | 0.30 |

**Notes.**

Abbreviations: Dp, Dermatophagoides pterronyssinus.

* p < 0.01, ** p < 0.001: Comparisons on sensitization between 3 districts.

Gray shadow: significant differences depending on PFAS status.

**Table 4  The variables affecting Pollen Food Allergy Syndrome (PFAS) in all patients.**

| Variables | Univariable | | | Multivariable | | |
|---|---|---|---|---|---|---|
| | OR | CI% 95 | p | OR | CI%95 | p |
| Male | 0.92 | 0.39–2.17 | 0.845 | | | |
| Aged >10years | 1.69 | 0.72–3.99 | 0.229 | | | |
| Seasonal AR | 1.41 | 0.59–3.36 | 0.442 | | | |
| Location | | | | | | |
| Busan | 1 | | | 1 | | |
| Fukuoka | 2.38 | 0.88–6.44 | 0.088 | 1.56 | 0.24–10.23 | 0.642 |
| Tochigi | 3.06 | 0.89–10.59 | 0.077 | 1.14 | 0.18–7.39 | 0.894 |
| Sensitization to pollen | | | | | | |
| Japanese cedar | 3.92 | 1.47–10.49 | 0.007 | 2.59 | 0.42–16.03 | 0.307 |
| Pine | 6.56 | 1.95–22.08 | 0.002 | 1.48 | 0.17–12.98 | 0.723 |
| Alder | 6.72 | 2.69–16.8 | <0.001 | 6.62 | 1.63–26.87 | 0.008 |
| Ragweed | 7.50 | 2.57–21.86 | <0.001 | 1.16 | 0.13–10.15 | 0.896 |
| Japanese Hop | 5.88 | 2.00–17.26 | 0.001 | 0.71 | 0.07–6.99 | 0.768 |

**Notes.**

Abbreviations: AR, allergic rhinitis.

Foods such as kiwi, peanuts, cashew nuts, banana, tomato, mango, pineapple, mandarin, melon, chestnuts, spinach, and eggplant were found to cause PFAS among children in Fukuoka (Table 5). Specifically, the number of children with PFAS caused by peach was higher in Busan and Tochigi than in Fukuoka, while kiwi and peanuts were the causative foods among children with PFAS at all sites.

**Table 5** Differences of subjects with Pollen Food Allergy Syndrome (PFAS) between three sites.

| Location | Busan | Fukuoka | Tochigi |
|---|---|---|---|
| Number | 7 | 14 | 6 |
| Male, (%) | 4 (57) | 7 (50) | 4 (67) |
| Median age (years, range) | 10.6 (7–15) | 9.5 (6–13) | 10.2 (7-15) |
| Causative food | | | |
| Peach* (%) | 4 (57) | | 2 (33) |
| Apple (%) | 2 (29) | | 1 (17) |
| Kiwi (%) | 1 (14) | 4 (29) | 2 (33) |
| Peanut (%) | 3 (43) | 1 (7) | 1 (17) |
| Walnut (%) | | 4 (29) | 2 (33) |
| Tomato (%) | | 1 (7) | 2 (33) |
| Cashew nuts (%) | | 2 (14) | |
| Banana (%) | | 2 (14) | |
| Mango (%) | | 1 (7) | |
| Pineapple (%) | | 1 (7) | |
| Mandarin (%) | | 1 (7) | |
| Melon (%) | | 1 (7) | |
| Chestnut (%) | | 1 (7) | |
| Spinach (%) | | 1 (7) | |
| Eggplant (%) | | 1 (7) | |

**Notes.**
*$p < 0.05$: Comparisons on causative food between 3 districts.

## DISCUSSION

In the present study, we demonstrated that alder pollen sensitization was associated with PFAS among children with allergies in three regions of South Korea and Japan. Alder trees are distributed across the Korean Peninsula and all areas of Japan. In Busan, alder pollen was the most common pollen after pine pollen and it had a 71% sensitization rate among patients with PFAS. Similarly, in Fukuoka and Tochigi, although alder pollen was rare, patients with PFAS had a high alder pollen sensitization rate.

In Europe, birch pollen rather than alder pollen is a common cause of PFAS (*Biedermann et al., 2019*). Birch vegetation is prevalent in Busan but not in Fukuoka (*Osawa et al., 2020*). This corresponds to the annual airborne pollen counts that showed only a small proportion of birch pollen in Busan and Tochigi, while no birch pollen was found in Fukuoka. In South Korea and Japan, although there is some birch pollen dispersal, sensitization to pollen occurs from both the Betulaceae (alder and birch) and Fagaceae families (oak, chestnut, and walnut), which could comprise the birch homologous group causing PFAS (*Masumoto et al., 2018*), as in Europe (*Pedrosa et al., 2020*).

The birch allergen Bet v 1 (also known as PR-10), a component of apple and peanut, was also found to be associated with peach, which is a common causative food for PFAS among children in Busan and Tochigi. Unfortunately, Betv1-IgE measurements were not investigated in this study. In Fukuoka, however, peach was not the causative food item of PFAS, yet a high sensitization to birch, alder, and peach was observed. Previous studies have

revealed that the average yearly consumption of peaches in Busan, Tochigi, and Fukuoka is 16, 1.9, and 0.8 per person, and that of apples is 27, 19, and 15 per person (*Statista Research Department, 2022*; *Regional Container, 2022*). Therefore, the lack of peach and apple-related PFAS detection in Fukuoka may be attributed to the low peach and apple consumption in the region.

Patients with PFAS in Fukuoka had a higher sensitization rate to ragweed and Japanese hop than those in Tochigi and Busan. The ragweed pollen allergens include Amb a 6 (known as LTP), which is a component of kiwi, melon, peanut, walnut, apple, and peach, and Amb a 8, which belongs to the Profilin family and is found in kiwi, melon, pineapple, peanut, apple, banana, and peach (*Carlson & Coop, 2019*). Japanese hop is a major allergenic airborne pollen in Korea, and it was not significantly associated with ragweed pollen sensitization (*Park et al., 2001*). Among children in Fukuoka, Japanese hop was an important allergenic airborne pollen causing PFAS (*Jang et al., 2021*).

Despite the high pollen counts in Busan, patients were not as sensitized to Japanese hop as in Fukuoka. Furthermore, in Busan and Fukuoka, the quantity of grass and weed pollen collected in each region was significantly smaller than the quantity of tree pollen, and there was no significant annual change in the pollen counts as in the case of Japanese cedar pollen (*Kishikawa et al., 2019*; *Sung et al., 2014*). Importantly, the Japanese and Korean methods of assessing airborne pollen counts differ; therefore, we could not compare the absolute value of the airborne pollen counts between the two countries. The pollen of Japanese hop is small and light; thus, it may have been underestimated by the Durham sampler, which is the gravity method in Fukuoka. However, pollen counts obtained by Rotorod and Durham sampler methods were previously reported to be correlated (*Sun et al., 2017*).

The airborne pollen sensitization rates may vary based on the amount of airborne pollen scattered and the characteristics of children with the allergy. Japanese cedar pollen was previously demonstrated to cross-react with tomato pollen (*Kondo et al., 2002*). In our study, the sensitization rate to tomato was significantly higher among individuals in Japan. This may explain the occurrence of PFAS associated with tomato among children in Fukuoka and Tochigi. Pine pollen has not yet been molecularly analyzed, but a previous study has shown immunological cross-reactivity between airborne pollen proteins and pine nuts, which are not consumed in Japan (*Senna et al., 2000*). In Fukuoka, pine pollen might have an allergenic component that causes PFAS. Therefore, further investigation on the allergenicity of pine and Japanese hop pollen should be performed to better understand their molecular backgrounds.

This study has several limitations. First, specific IgE tests for different types of airborne pollen after pollen dispersal were not conducted. If tests for various types of grass and weed pollen were performed after pollen dispersal, their association with PFAS could be further investigated. Second, only few patients from Tochigi were included in the analysis. Third, in this study, pollen count data were obtained from a single year. The amount of pollen dispersal varies yearly depending on the annual temperature and humidity, thus, limiting our evaluation of the relationship between pollen counts and sensitization. Finally, the proportion of patients with BA and FA were higher in Fukuoka and Tochigi than in Busan,

which supports previous findings on the higher prevalence of BA and FA in Japanese children than in Korean children (*Lai et al., 2009*; *Kim et al., 2015*).

## CONCLUSIONS

Despite a small proportion or none of birch pollen was observed, alder pollen sensitization was associated with PFAS among children at Pusan in South Korea and at Tochigi and Fukuoka in Japan. Ragweed and Japanese hop pollen sensitization were found to cause PFAS, particularly among children at Fukuoka in southern Japan. The characteristics of PFAS varied according to local vegetation, pollen size and weight, eating habits, and comorbid allergic diseases. Further investigation is required to assess the molecular background of local airborne pollen and food items causing PFAS.

## ACKNOWLEDGEMENTS

We thank Professor Jae-won Oh of the Hanyang University Guri Hospital, Dr. Kei Satou of the Satou Respiratory Clinic, and Ms. Eiko Kotoh of the National Hospital Organization Fukuoka National Hospital for providing airborne pollen data.

### Funding
The authors received no funding for this work.

### Competing Interests
The authors declare there are no competing interests.

### Author Contributions
- Yoonha Hwang conceived and designed the experiments, performed the experiments, analyzed the data, prepared figures and/or tables, authored or reviewed drafts of the article, and approved the final draft.
- Chikako Motomura conceived and designed the experiments, performed the experiments, analyzed the data, prepared figures and/or tables, authored or reviewed drafts of the article, and approved the final draft.
- Hironobu Fukuda performed the experiments, authored or reviewed drafts of the article, and approved the final draft.
- Reiko Kishikawa performed the experiments, authored or reviewed drafts of the article, and approved the final draft.
- Naoto Watanabe conceived and designed the experiments, authored or reviewed drafts of the article, and approved the final draft.
- Shigemi Yoshihara conceived and designed the experiments, authored or reviewed drafts of the article, and approved the final draft.

## Human Ethics

The following information was supplied relating to ethical approvals (*i.e.*, approving body and any reference numbers):

The Research Ethics Committee of Busan St. Mary's Hospital (BSM 2016-14), Fukuoka National Hospital (F28-3) and Dokkyo Medical University Hospital approved the study (29031).

## Data Availability

The raw data is available in the Supplemental File.

## Supplemental Information

Supplemental information for this article can be found online at http://dx.doi.org/10.7717/peerj.14243#supplemental-information.

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
