# Peer review of "Relationship among airborne pollen, sensitization, and pollen food allergy syndrome in Asian allergic children"

_PeerJ, doi:10.7717/peerj.14243_

## Round 0.1 · original submission · Major Revisions

Please provide requested revisions by the reviewers or a detailed point-by-point rebuttal.

Reviewer 1 ·

Basic reporting

The text is generally well written. I miss the tables with questionnaires used (PFAS questionnaire).
Concerning the fact that this is cross sectional study and the goals of the study are to analyze airborne pollen counts, IgE sensitization rates, and PFAS incidence among children with allergies in South Korea and Japan the conclusion of the study doesn't entirely match the goals. It would be better to conclude on the data received and then to add the specific finding.

Experimental design

This is by methodology a cross section study - it would be nice to emphasize it and answer to the question is it epidemiological study or not. If it is, what percentage of population did this study included. How did you chose your patients?
The data about asthma, atopic dermatitis and PFAS are redrown only from questionnaires?

Validity of the findings

It is an interesting finding that all children with sensibilization to alder with PFAS are also sensitized to birch pollen. Did you do only specific IgE to birch or maybe to Bet v1?

Reviewer 2 ·

Basic reporting

no comment

Experimental design

Please explain the criteria of the recruitment. What are the limiting factors of enrollment?

Please describe the distance between Tochigi and Gunma pollen counting station.

Please explain if allergic diseases, including PFAS have been confirmed by the physician or if these data are from reports from the parents/patients.

Validity of the findings

Please explain what % of the subjects were enrolled from the recruitment and what % of the patients from your clinic were enrolled.

Are there any differences in the IgE level before and after the pollination season?

PFAS to apples is relatively low. Is the consumption of apples lower than peaches in these areas?

Are there any differences in the incidence of PFAS between before and after adolescence?

Additional comments

The comparison of PFAS among children in different regions is interesting.

---

## Round 0.2 · accepted · Accept

Your manuscript is ready for publication.

Reviewer 1 ·

Basic reporting

No comment.

Experimental design

No comment.

Validity of the findings

No comment.